# When Parallel Schools of Thought Fail to Converge: The Case of Cost Overruns in Project Management

**Aaron Anil Chadee** [1,*] , **Xsitaaz Twinkle Chadee** [2] , **Indrajit Ray** [1] , **Abrahams Mwasha** [1] and **Hector Hugh Martin** [3]

1   Department of Civil and Environmental Engineering, The University of the West Indies, St. Augustine, Trinidad and Tobago; indrajit.ray@sta.uwi.edu (I.R.); Abrahams.Mwasha@sta.uwi.edu (A.M.)
2   Department of Physics, The University of the West Indies, St. Augustine, Trinidad and Tobago; Xsitaaz.Chadee@sta.uwi.edu
3   School of Civil Engineering and Built Environment, Liverpool John Moores University, Liverpool 72046, UK; H.H.Martin@ljmu.ac.uk
*   Correspondence: aaronchadee@hotmail.com

**Abstract:** This paper investigates the conflicting and contrasting views between two prominent schools of thought (SOT), the conventional project management empirical school and the incoming behavioural and psychological sciences, to explain the cost overrun phenomenon in transportation infrastructure projects. Although theories within these SOTs seem to coexist and are used interchangeably, there exists a widening knowledge gap that leads to conflicting and contrasting ideological views. In this paper, we contend that there is a lack of a cross-fertilisation mechanism to fuse and co-join contemporary theories on cost overruns. This has led to the encapsulation and fragmented adhocracy in theory building. Utilising a critical review approach, this study proposes the concepts of ideological distancing and encapsulation between "empiricism" and "behavioural" SOTs as the focus of analyses for understanding the root causes of cost overruns in project studies. The study showed that the extant debate on cost overruns is limited and divergent, effectively contributing to the problem of continued expansion and non-convergence of theories that maintain parallel identities. This creates a space for inquiry, reflecting, theorising, and debating for the convergence of SOTs on cost overrun research and theories on what can be deemed project knowledge. This paper contributes to extant project studies by identifying the need for convergence and co-joining theories with different epistemes through causal linkages. Consequently, this will improve the public sector's infrastructure policymaking by exposing the theoretical limitations of the current ad hoc manner and application of project management concepts based on the project professionals' bounded decision-making rationalities.

**Keywords:** cost overruns; ideological distancing; encapsulation; empiricism; behavioural; infrastructure; public sector; project management; mega projects

## 1. Introduction

The new millennium (2000) heralded multiple breakthroughs in the field of project management (PM). In the 1950s, it began as an engineering and optimisation tool, and has grown by accommodating a multiplicity of disciplines in an ad hoc manner [1]. In its present form, PM is taking the shape of a research field, with opportunities to bring different disciplines to concentrate on a specific phenomenon of study, specifically, projects [2]. However, advancements in both practice and theory have made limited strides to consistently improve cost performance of projects. Large projects and mega-projects have been confronted with the recurring unsolved problem of cost overruns [3]. The wide publicity of the misuse of taxpayers' money has provoked outcry, placing the topic as a central issue for reflection, critique, and theory building. This tumult has engaged the attention of the two dominant disciplines in PM, which view the problem from different epistemic settings.

As recent as 2019, contemporary PM has been consumed in constructive criticism between the advocates of two leading ideologies, the conventional empirical (technical) and the evolving behavioural social sciences schools of thought (SOT). This tension began to unfold at the beginning of the new millennium, in 2002, from the publication of the article "Underestimating costs in public works: error or lie" [4]. This article sparked a controversy within the engineering community by using some conspicuous words, "Error or Lie", as root causes of cost overruns in the context of major transportation projects. "Error", referring to optimism or delusion in cost estimation, is based on psychological and behavioural sciences, while "lie", or strategic misrepresentation of costs by practitioners, is based on a conventional, mainstream empirical sciences field. The ideological distance between these two disciplines further increased in 2018, when the advocates [5] of mainstream empirical SOT decided to critically discredit and reject the behavioural attributions of the advocates of the social sciences school [6]. This unfolding debate on the root causes of cost overruns requires further investigation to determine whether different epistemologies can be co-joined within the PM frame.

In the process of diversifying its base, PM began to take on a broader perspective of activities and evolved into an organised field of research, or a discipline, in itself. In this endeavour, its theoretical constructs left behind openings and trails [7,8]. Tensions, conflicts, and debates surrounding the nature, understanding, interpretation, and application of theories and theoretical concepts were inevitable. This paper investigates the causes of ideological distancing between mainstream conventional PM and the incoming school of behavioural and psychological science, the limitations of the theories and the current diverging gap. This ideological confrontation is addressed through the following two research questions:

1.  What issues are contributing to the ideological divide in cost overrun research in PM?
2.  How can institutionalised diversity within PM fields contribute to the narrowing, rather than expansion, of theoretical gaps in PM?

Whereas past studies concentrated on the history of the two disciplines independently, this paper examines the contrasting changes in their parallel upbringing and their relationship with one another, and discusses how interdisciplinary formulations evolved, as well as the mechanisms behind those relationships [9]. This is quite evident from the writings of Flyvbjerg [6] and Love and Ahiaga-Dagbui [5]. A comparative relationship was possible, and particular attention was paid to how the two disciplines have conceptualised projects and cost management as a vehicle to foster new processes and theory. Before pursuing trajectories as contrasting contributors to solve long-standing problems of cost overruns and benefit shortfalls, project management started as a relatively integrated field of research. This period is reviewed, including the emergence of two contrasting ideologies built on diverse epistemic cultures, which would later influence the disciplines' parallel lanes.

## 2. Literature Review

The literature review was divided into three categories to capture the coherent trend across a plural oriented, multi-faceted, and disjointed discipline. The concepts of diversity and fragmentation, institutions and specialisation, and ideological distancing and encapsulation are reviewed herein to show the lack of theoretical convergence in PM.

### 2.1. Diversity and Fragmentation

In the pursuit of expanding its domain, PM acquired a multi-faceted, diverse, and ad hoc assembly of disciplinary concepts, all drawn from a wide variety of traditions and epistemic values [1]. Mintzberg [10] produced what he termed "a strategy formulation" of schools of thought, and a comprehensive list identifying seven traditions or schools of thought which controlled the operations of PM: optimisation, factor, contingency, behaviour, governance, relationship, and decision schools [11].

The earliest model was the optimisation school of thought, which was practiced during the aerospace era of the 1950s, and is still being used today. It is based on the

well-established conventional empirical model of engineering and applied mathematics, with an emphasis on meeting deadlines, staying under budget, and adhering to specifications [12,13]. This school (tradition) was developed to answer how projects are planned and managed. It brought about rationalistic, logic-based planning, formal processes, and analytical techniques to predict future outcomes. According to [14], there was a need to positively impact the bureaucratic stakeholders of these sophisticated and high-end projects by careful upfront planning to select the optimal technologies, detailed scheduling of project activities, and prearranged integration of components in the final system.

A decade later, in the 1960s, limitations were revealed in the optimisation school. While the intellectual contribution of an individual is essential to increasing performance and productivity [15], engaging the subjectivity of the human element proved to be a necessary ingredient to increase productivity [16]. Thus, the discipline of PM expanded to include a more active role for the project manager. This gave rise to a new line of thinking and theorising through the factor tradition. The emerging factor tradition recognised that managing a project was not a sufficient guarantee of project success. This tradition has evolved to answer the question of what determines project success. Hence, there is a need for PM to address the issues that limit successful project outcomes by identifying the key factors.

The diversity and the inclusion of a wide range of concepts both in PM and individual projects brought with them misunderstandings, conflicts, and the tendency for planners to drift apart, thus creating a communication gap. By the 1970s, PM expanded to address the important question of why projects differ [17,18]. This included the contingency school of thought, which broadened the concept of management integration with intellectual integration [19]. At around the same time, emerging from the question of why projects differ was the motivation to search for its root causes and fundamental principles. These studies are grouped under the tradition of organisational theory and classified as the behaviour school to address the key question of project evolution.

In the 1980s, PM began to provide an avenue for cross-fertilisation of its knowledge base from social constructs into economics [20]. The need for a governance paradigm was essential to allow the flow of knowledge on how projects were to be governed. Concurrently, PM naturally extends to transition, and follows market expansion and globalisation [21]. The key question of this tradition was to address how project networks are formed and managed. This tradition focusses on stakeholders' relations and their linkages between theory and practice [22]. Finally, at the end of the last millennium, a concerted attempt was made to address the intrusion of politics in influencing project decision-making processes. This latest decision tradition investigated why projects were instigated, and the causes for their continued existence [23]. These schools expanded the PM domain beyond the proposals presented by projectification [21,24] and projects as temporary organisations [25].

Each tradition is rooted in its own custom, leading to the emergence of several theories held together without a strong set of common core principles and ideas [26]. The theory of projects is limited if it is dependent solely on empirical insights and needs to be driven by a particular theoretical perspective [27]. A number of characteristic features were attached to the various schools: (1) project definitions differed, (2) project management was swayed by different opinions, (3) research was influenced by different key questions, (4) there was pluralism, and (5) a diverse range of disciplines entered to influence PM.

### 2.2. Institutions and Specialisation

The forerunner to the arrival of many schools or traditions is the apparent streamlining and specialisation of each individual school and the separate institutionalisation promoting individual concepts. Through the process of specialisation and fragmentation, several subdisciplines evolved [28,29] with each fragment purporting to identify some epistemological links to cushion its institutional framing. The subdisciplines co-evolved to become attached to particular research questions, associations, journals, and universities [30]. Though scholarly publications are demanding and time consuming, there is

little incentive to integrate bodies of knowledge if research specialisation is reflected and supported by institutional specialisation, as in the case of management subdisciplines [1]. In this process, an inherent tension develops between competing entities that strive for their own space and expression, leaving little room for integration [1,26,31–33].

While specialisation is a core incentive to probe deeper for alternative solutions in a particular field, there is a tendency to concentrate on separate and isolated subdisciplines. This furthers the drift in PM diversity, and shows how specialisation promotes endogenous theory-building [34] which leaves little room to address complex societal problems [35–39]. This has the effect of discouraging researchers from pursuing large-scale ventures where there is a need for collaboration across disciplines [40,41]. There is also ambiguity surrounding research terminologies, leading to the emergence and co-existence of multiple paradigms [31]. This ad hoc fragmentation serves no useful aim in the configuration of more fundamental theoretical issues [35,39].

### 2.3. Ideological Distancing and Encapsulation

The expansion of PM into a number of traditions or schools of thought introduced a diverse and multi-faceted framework that was held together in an ad hoc manner [10]. Each school is preoccupied with its own notion of what is a project, what constitutes the central focus, and its ability to handle specific questions. Thus, projects within different disciplines, belonging to separate knowledge domains, tend to project their own "thought world", and not only know different things, but interpret the same thing differently [42,43]. This can affect and modify the criteria for a system's assumptions, design, and positive changes [44].

A basic characteristic of each school is its ability to portray a unique project definition. This provided the means to grow into a specialised field, entrenching itself with its distinct modus operandi. Reflecting on the number of schools within PM, there is reasonable evidence suggesting that several theories will emerge to explain PM. These theories rely on diverse theoretical foundations, spanning a spectrum from applied mathematics to psychology and political science [2].

It is challenging to bring logical coherence to the varied theoretical alignments existing within PM. The ability to effectively design and integrate a comprehensive program in academic research to fit a wide range of knowledge is a daunting task [45–48]. Team members from contrasting disciplines can fall prey to culture clashes, as individuals try to maintain their sometimes-conflicting norms and values ingrained in their epistemic communities [49,50]. In the pursuit of holding together a multi-faceted PM discipline, an ideological divide surfaced. This is as communities have their own methods of defining problems, sourcing, and interpreting data, even if there is a common interest in the subject at hand.

Rouleau and Séguin [51] commented on the development of this parallel problem which leads to ideological distancing, where self-referential ideologies may, and sometimes wilfully, side-line important perspectives and interpretations [30]. The promotion of such ideologies can lead to encapsulation—the limited sharing of ideas and concepts from related disciplines bearing similar ideologies [30].

Such ideological distancing and encapsulation of PM notwithstanding, there exist different strains of theories, some of which were fundamentally conceived with foundational intellectual roots that can evolve into a unique model capable of describing universal events, or at least endeavour to unfold generic traits. Thus, the original "mainstream projects" carried the argument as far as certainty, and with the extension of PM, some ideologies introduced uncertainty, a necessary factor to be included in the new dispensation to further extend aleatory and episteme profiling [52].

### 2.4. The Ideological "War of Words"

Considering the large number of high-profile scholars engaged in investigating the cause of projects cost overruns [53,54], and through this ideological "war of words", a

comparison of equal status can be made to show the unfolding debate between these prominent scholars. This was prompted by Flyvbjerg et al. [6] (p. 186) who comments:

> *"it conveys a deterministic type of thinking we would have thought extinct in the academia after the probabilistic revolution has shown that nothing is deterministic in nature. The claim that we must know "what will occur" to make an estimate is akin to insisting on understanding the world in terms of Newtonian physics after quantum mechanics, something you would not get away with in physics."*

In science, two theories emerge to explain the theoretical fundamentals and, consequently, divide the world of physics into two halves—the deterministic and the probabilistic—the former headed by the famous Albert Einstein and the latter from the followers of quantum theory, a fundamental mathematical model in probabilities. Science took centre stage in the last century of the last millennium, when the deterministic, led by Einstein, challenged the probabilistic of quantum theory fame in a prolonged war of words to justify which mathematical model was superior in explaining the universality of physics. In the 21st century, a similar paradigm shift arrived in project management, noted as Kuhnian's paradigm shifts in the fields of behavioural science and project management [6].

The contrasting development of conventional empirical evidence model and behavioural social science research on root causes of cost overruns on major transportation projects, and by extension major projects, serve to challenge existing theory and motivate new ideas and thinking [55]. Although seeking solutions to the same problem, there exists a parallel divergent path that inhibits the sharing of domains, ideas, and concepts of mutual interest [22,36,56]. As time progressed, it became evident that these neighbouring disciplines failed to recognise each other's contribution to their shared research agenda [4,5,7,14,19,57], despite the increasing clamour for convergency by many [58–61]. This creates an ideal phenomenon of interest, and a contemporary case through which to examine and extract theory about the tensions and facilitators of cross integrating neighbouring disciplines in PM.

From the turn of the new century, the critique between the leading scholars of PM continued, with the debate focusing on large monetary injections in mega projects [3]. As projects moved from the mega to the tera age, costs could run in the billions of dollars. Any underestimation in a project can lead to an appreciable amount of cost overrun, which has become a central focus in PM's debate. As infrastructure works and large-scale projects become prevalent and are undertaken by governments and private contractors, public funds (taxpayer money) attract the attention of the entire citizenry.

## 3. Method

A qualitative method was used to develop new theoretical considerations based on single and multi-case studies combined into the singular phenomenon of project cost overruns. Case studies are an excellent source for detailed empirical descriptions of a specific instance of a phenomenon that emanates from various data sources [62]. Case studies are an effective method for analysing complex and less well-understood processes [62,63], highlighting the mechanisms and constraints that exist across research disciplines. With this in mind, a critical qualitative case study approach was used to focus on the two main contrasting disciplines, which provided topical interests and concepts to demonstrate the fundamental fault lines and divergences of which theoretical stance provides more profound insights into the intellectual root derivative of cost overruns. The theoretical debate was critiqued based on ideological distancing and encapsulation, and proffered future studies for a mechanism for convergence for both SOTs. Both the selection and analysis of the respective studies are informed by prior research on cost overrun and associated benefit underperformance, including intellectual root derivatives, as detailed below. As a result, this study builds theory through case studies, a research strategy that entails selecting one or more cases to serve as a platform for developing a workable theoretical formulation based on strong empirical evidence [64].

Considering the examples set by other qualitative reviews in research evolution (e.g., for international business see [65], for projects see [66]), the study provides the need for interesting original models in PM theory's convergence and continuity. This case study is based on excerpts from a leading journal. First, the contenders to this project cost overrun debate are detailed to show, among other things, the magnitude, gravity, scholarly inputs, and global interest the cost overrun phenomena has attracted. The editors supervised the arguments and guided both sides to the opportunities of rejoinders. Rejoinders are helpful toolkits, as they serve to advance or clarify issues associated with the individual sides and to express how the contributions can amalgamate to further the overall interest in PM and PM theoretical bases. A leading source of data was a review of the literature, and the subsequent unfolding of a tension introduced in a 2002 publication [67]. Instead of following a co-citation or network analysis [56], a qualitative rather than a quantitative approach was adopted to unearth important changes in knowledge integration of core ideas over a long duration and the root mechanism hindering or facilitating continuity. Following this trend, an inductive and multi-case approach was employed to advance analytical generalisation and theorising to lead to future research [63,64].

The preparation of the literature review entailed consulting various sources, including seminal works, theoretical articles, scholarly journals, periodicals, and research designs. These sources were largely extracted from academic databases, including (but not limited to) ProQuest, Elsevier, Sage, EBSCOhost, ebrary, Business Source Complete, RefWorks, IEEE Xplore, Science Direct, and SpringerLink. Google Scholar and Google Books were also used to search for peer-reviewed and scholarly journals and seminal works. These databases contain records of keywords and abstracts from 1970 to 2020. The compilation of literature review data included other sources, such as archival data and observations from workshops and conferences, to assist data triangulation and cross-validation [62,68]. The key terms used to search within these databases were: project management theory, mega projects, root causes, and cost overrun.

Two parameters were employed in the screening process. Firstly, the keywords authors used to describe their works provided the initial acquisition of scholarly works. Secondly, abstracts from the first screening process were subsequently reviewed to determine their relevance within the scope of this research. This review process also included referenced published bibliometric and systematic literature reviews within publications; these reviews were also screened to trace the evolution of each tradition and formulate a consensus around specific methods to determine new concepts and ideas in each discipline. Further, an analysis of potential cross integration pertaining to the studies was conducted by screening recent issues of leading journals and capturing editorial and reviewing boards' inputs as indicators to substantiate the claim that the major studies were viewed from a prominent position in the intellectual PM society. The temporal framing [69–71] provided a pathway for the narrative reconstruction of the contrasting development of the two behavioural and empirical sciences models, as prior reviews have acknowledged that engineering evidence science and behavioural science have contrasting roots [72]. The focus was to identify important categories in the evolution of these project management studies that can clarify hindrances and propel the continuity of theory. Some notable parallel systems identified from the analysis and can be included into the three groupings: (1) the emergence stage (from contrasting roots); (2) ideological distancing; and (3) the failure to converge (the non-linearity model).

## 4. Case Study

### 4.1. The Emergence of Engineering Evidence Science and Social Behavioural Science

The two disciplines of engineering science and social behavioural science are frequently at odds in PM with different underlying theories (for example, Optimisation vs. Optimism) and topics of interest (cost-overrun and benefit shortfalls). Due to their parallel concepts and diverse epistemic cultures, the two disciplines have failed to recognise each other in terms of PM validity. This section will analyse parallel concepts with diverse

epistemic cultures and the claims to explain the root intellectual causes of the conflict and the resulting cost-overrun in PM.

The most popular and cited publication in cost overruns on transport projects [4] was recognised as being one of the highest profiled studies of cost overruns [73] among the top five most-cited research of the Journal of the American Planning Association [5]. The publication by Flyvbjerg, Holm, and Buhl [4] sought to further the causes of behavioural science as an incoming and second discipline alongside the prevailing established one of engineering evidence science, in the pursuit of solving the long-standing problem in PM known as cost overrun. As behavioural science stems from a different epistemic root which is distinctly diverse from conventional science, the theoretical and conceptual rationale brings a completely different way of examining cost overrun and project management. It is not unusual for critical analysis to create friction and even confrontation, especially when the diversity of thought and arrogance are intertwined in the agenda. The seed of discontent was laid down by two simple words separated by the two-letter word "or". It was extracted from the title of the publication that read as follows: "Underestimating costs in public works projects: "Error or Lie". "Error" is interpreted as originating from the traditional mainstream school of thought; "Lie" is an alternative interpretation proposed by the behavioural science discipline. "Or", with all its simplicity, means a divide. Herein lies the clue for ideological distancing. Sixteen years later in 2018, the theoretical gap between the dominant schools widened.

This paper examines the well-known unsolved problem of cost overrun in projects, and in particular large-scale infrastructure projects, which are undertaken on a global scale by all countries. The general consensus is that cost underestimation, also more commonly referred to as cost overrun, is prevalent, e.g., [20,74–81]. While it is acknowledged that cost overruns are a pervasive problem, the solutions presented are limited, and have attracted substantial attention in the media, with stakeholders, including the general public and academic scholars. At this point, it is not certain how cost overrun is defined, why it happens, or how to best circumvent it [6].

### 4.2. Ideological Distancing: On De-Bunking "Fake News" in a Post-Truth Era

The editors of Transportation Research Part A: Policy and Practice (2018) have taken a keen interest in the Love and Ahiaga-Dagbui [5] article titled, "De-bunking 'fake news' in a post-truth era: the plausible untruths of cost underestimation in transport infrastructure projects". The paper deals with cost underestimation, commonly known as cost overrun, in infrastructure projects, with particular reference to large scale or mega-projects. It has long been known that a major unsolved problem in PM is cost overrun. Deviation due to estimation in contract price, which is tallied in terms of billions of dollars, can lead to an appreciable sum sufficient to interfere with a country's national budget. By extension, there is clamour, especially from the news media, searching for sensational headlines, such as corruption, that attract wider patronising market viewership. The presentation was not fashioned in the normal academic style, and the title of the first publication suggested a confrontational approach between schools of thought.

There are two main schools of thought in PM that are currently engaged in finding solutions to this problem [2]. They are the traditional mainstream evidence school, which was in operation from the inception of PM, and the incoming school, behavioural science, which has a different epistemic background. Part of the title headline by Love and Ahiaga-Dagbui [5] refers to "de-bunking fake news," which suggests that Love and Ahiaga-Dagbui [5], the advocates of mainstream evidence science, are challenging the followers of the behavioural school. The leading academics of the opposite school are listed in the journal in the following paragraph:

*"This overly critical paper received a strong comment by another group of distinguished scholars: Bent Flyvbjerg, Atif Ansar, Alexander Budzier, Soren Buhl, Chantal Cantarelli, Massimo Garbuio, Carsten Glenting, Mette Skamris Holm, Dan Lovallo, Daniel Lunn, Eric Molin, Arne Ronnest, Allison Stewart and Bert van Wee, which was also published*

*in 2018 with the title, "Five things you should know about cost overrun.""* (ibid, pp. 174–190)

Notwithstanding this strong objection, it was Flyvbjerg et al.'s [6] comments that instigated this dispute by retorting:

> *"Love and Ahiaga-Dagbui [5] may dislike research results like these because they identify members of their profession as unethical."* (p. 184)

"Members of their profession" identifies a whole school of scholars who may be guilty of malpractice. Incidentally, engineering science and profession have been dominating the PM's practice for over 70 years [4]. Failing to recognise each other's contributions, both teams entered the intellectual arena to defend, "de-bunk", and champion the cause of their own status quo.

To the average taxpayer, cost overrun on public sector projects means the loss of money, and with the spin from the news media, it is seen as a form of illegal gains. The behavioural school was on target, pushing a line of thinking that focused the attention of the professional planner as the main culprit. The new millennium brought with it projects worth trillions of dollars. As the population was still growing accustomed to counting in the billions of dollars; the tera age had dawned, bringing with it a monetary dimension that the mind was not used to, large enough to confuse and frighten the ordinary taxpayer, the main stakeholder.

It is still an open field at this point, as there is no universally accepted explanation for cost overrun [5]. While there has been some improvement due to the constant search by mainstream technical input, e.g., [77,82–84], the mitigation and containment strategies employed to yield conclusive results fell short of the intended goal. This persistent failure by mainstream models did not go unnoticed, but rather attracted the attention of others, especially those who were from different disciplines in search of opportunities [4].

> *"It is striking that this long-standing pattern (of cost overruns), which appears to prevail worldwide, continues unabated despite major improvements in technical capacity for cost estimation–suggestion that its causes lie primarily in the realm of politics rather those of engineering or accounting."* [85] (p. 221)

Although the 2002 article by Flyvbjerg initially lay down the foundation for conflict, it was the Love and Ahiaga-Dagbui [5] article that propelled the debate further, gaining momentum at every turn and, surprisingly, drawing in more scholars to either side. It is evident that a main ingredient in this debate extends further than the exchange of ideas. The fact that a step was taken to appeal to competing institutions to defend the status quo shows the keen interests and academic ranking attached to this debate. Under the supervision of Love and Ahiaga-Dagbui [5], assembled scholars over the period of preparation, from 2002 to 2019, included Ika [54], Zhou, Edwards, Irani and Sing [81], Söderlund [72] and Siemiatycki [73]. The readership of the debate extended through universities from Australia, Europe, Canada, the United States of America, and central Asian countries. This wide encashment qualifies this continuous friction over many years as a war of words in the new millennium. This is best summarised in the concluding paragraph of [5] as follows:

> *"If cost underestimation is to be effectively addressed and good decisions at the outset of a project are to be made in the future, then there is a need for these estimates to be based on reality and not on delusion or falsehoods. Weakening the link between evidence and decisions not only jeopardises the quality of transport policymaking, it threatens the entire enterprise of scientific research."* (p. 366)

The rising popularity and the continuous surge forward since 2002 began to position behavioural science with an advantage over its long-standing mainstream competitor. The proponents of mainstream science expressed a sign of decline and, after contemplating the issue for more than a decade, decided to act in a retaliatory manner to compensate for the adverse friction generated since 2002. Apart from attempting to de-bunk the other side, the

publication addresses, among other things, a philosophic and theoretical concept that will arise when dealing with a multi-faceted discipline as project management [10,86].

The scholars of the behavioural school of thought, embedded in social sciences, were echoing the call of their predecessors of the 1970s when Prospect Theory [87] produced a new roadmap along an inconvenience path, embodied in the unusual non-scientific frame of social, behavioural, and psychological diction. To slip in this new tradition in a field that was reserved for science and engineering was considered at best, provocative. The mantra of discourse emanating from social science was too speculative, but sufficient, to run a parallel institution within PM—long known for accepting diverse disciplines in an ad hoc manner [1,26].

It is not unusual for scholars to embark on a piercing critical analysis of each other's work, but it is rather strange to insinuate a line of confrontation and extend an open arm to stakeholders and policymakers to revert to the original format. The seed of discontent was sown some sixteen years ago and had a long gestation with little or no criticism until the arrival of Love and Ahiaga-Dagbui [5], who aggressively confronted the problem by pointing out a series of myths in the Flyvbjerg, Holm, and Buhl [4] article. Of fundamental interest in theory building is myth four, which was stated as follows:

*" . . . . . . However, their bifurcation of the cost underestimation problem into error or lie presents the reader with a false dichotomy, an either/or choice that is practically invalid when juxtaposed with the real-world nature of procuring large infrastructure assets . . . . . . This false dichotomy forces the reader to reject complexity in complex decisions and focus on only the two extremes presented, with the assumption that no middle options are available."* (p. 365)

### 4.3. Failure to Converge—The Behavioural School of Thought

The following is a critical reply to the Love and Ahiaga-Dagbui [5] paper, which debated the views of mainstream technical science, which is founded on the conventional wisdom of on time, within budget, and according to specification [12,13]. To defend the parallel running behavioural school, Flyvbjerg sourced a team of international scholars to challenge the validity of the mainstream technical doctrine by a style that is seldom witnessed in project management. The stimulus to defend this cause rests with the aggressive words of Love and Ahiaga-Dagbui [5], "to de-bunk fake news", and the clarion call to policymakers to abandon behavioural science.

Flyvbjerg, Ansar, Budzier, Buhl, Cantarelli, Garbuio, Glenting, Holm, Lovallo, and Lunn's reply [6] carried the simple and affirmative heading: "Five things you should know about cost overrun". It lays out an overview of good and bad practices in large capital investment projects, using Love and Ahiaga-Dagbui [5] works as a point of departure from good practices. Due to the confrontational nature of the debate, the accusation of cherry-picking data and statistical analysis were side-lined to give way to the discussion on the fundamental theoretical and empirical important derivatives obtainable from either side. Flyvbjerg proposed two main root causes, optimism bias and strategic misrepresentation.

Optimism bias is a psychological flaw describing managers' tendency to make decisions based on delusional optimism, rather than on a rational weighting of gains, losses, and probabilities [4]. Such optimism ignores actual distributional time and cost performance data of similar tasks, falling prey to the planning fallacy [23,88,89]. Managers overestimate benefits, underestimate costs and time, and involuntarily spin scenarios of success while overlooking the potential for mistakes and miscalculations [89]. Strategic misrepresentation was the second leading root cause for cost overrun. This is the deliberate misrepresentation of costs and benefits made to deceive others, the most basic explanations of lying exist [4].

This raises a fundamental question: How did behavioural science get into Project Management without imposing itself as another ad hoc infusion? [10,26]. This is a subject of open dispute on how two contrasting models of behavioural science and technical science, with different epistemic values, became enmeshed.

## 5. Discussion

The first issue identified is: should one school attempt to de-bunk another school, using terms such as "fake news", when both emanated from the same PM framework? There is a tendency for schools of thought to develop along parallel lines in this multi-faceted and ad hoc PM framework. Nonetheless, as long as research specialisation is represented and promoted by institutional specialisation, there is often little motivation to incorporate incoming disciplines. With various epistemic fundamentals, the schools have split by ideological distancing. As to "fake news", the proponents of the mainstream evidence school are overtly suspicious and critical of the behavioural school. This "fake news" lay the foundation for friction and confrontation. Bearing in mind the number of followers on both sides, and their academic ranking of scholars at the highest level, there was a need to seek solutions. However, this appeared to be only part of the problem, as expressed by Flyvbjerg et al. [6]:

> *For instance, they describe our research findings as "fake news", "myths" (no less than 15 times), "canards", "factoids", "flagrant", "rhetoric", "misinformation", and more. We are further accused of having "fooled many people" by having "been just as crafty as Machiavelli" as we "have feigned and dissembled information" "through our research."* (p. 358)

Commenting on the language style, Flyvbjerg et al. claimed to have welcomed the objections as "criticism is the main mechanism for securing high levels of validity and reliability in scholarship". However, Flyvbjerg et al. [6] brought to the fore that their contribution to PM was in jeopardy.

> *"As a factual observation, in our entire careers, we have never come across language in an academic journal like that used by Love and Ahiaga-Dagbui [5]. We suggest such language has no place in academic discourse."* (p. 175)

Contrasting insinuations and confrontational attitudes are invoked along the path to PM's theoretical expansion. Such provocations are not to be ignored and excluded in academic debates, where perceived offensive lines of inquiries are made in the quest for knowledge. In fact, these lines of inquiry should be included in an academic way to remove them from the mathematical intricacies that make it difficult for the overwhelming majority to contribute, including philosophers. This was the popular cry of philosophers and literary writers [90–92], who noted the limitations of philosophers, such as Wittgenstein, to disengage from the mathematical complexities, and who withdrew from the last science war of words.

The explanations to this myth carried through three concepts: a pseudo dichotomy in logic, contrasting definitions of cost overruns through institutional variation, and encapsulation.

### 5.1. Pseudo Dichotomy in Logic

A critical analytical leverage is the means to deconstruct management studies into a number of subdivisions. The philosophical works of Derrida have inspired many management scholars [93,94] to pursue this path. Derrida [95] was able to concentrate on human interaction as production of texts, and argues that there is nothing outside the text [96]. That is, in the search for meaning, we use the medium and properties of language to convey meaning. However, a property of language also precludes conveying meaning in its absolute form. Therefore, scholars adopt "signifiers", as quoted above, to convey meaning. The text translates hierarchical structures formulated in terms of binary dichotomies to facilitate a theoretical analysis. However, language within the text possesses elements of incompleteness and contradictions, and eventually fails as meaning moves from one "signifier" to another without converging towards a definite form.

A theoretical construction of fundamental importance is the mechanism to deconstruct management profiling into a dichotomous arrangement, where the extremities form the main criteria central to a conclusive decision. The negative feedback to this approach

creates a litany of problems, one of which is exemplified from the extraction of the primary myth of the Love and Ahiaga-Dagbui [5] paper. They contend that this subdivision into two parts presents a pseudo dichotomy; while deconstructing management studies along the lines of Derrida's binary dichotomy, a weak theoretical construct is in the making. Chosen words do not stand up to PM expectations, and do not provide a convincing complimentary description of dichotomous arrangement.

The discussion of subdivision went beyond dichotomy, from comparing and contrasting binary items to a wider realm of analysis to include the rhetorical device "fallacy of the Excluded Middle". Love and Ahiaga-Dagbui [5] assert Flyvbjerg, Holm, and Buhl's "error or lie" [4] is a false dichotomy and misleading diagnosis, as there are many other explanations for the cost underestimation problem.

The fallacy of the Excluded Middle points to the third law of logic, laid down by Aristotle (Metaphysics, Book iv, Part 7). It is known as "the Law of the excluded Third", in which, when a statement is premised as either true or false, there is no middle option. The error or lie framing leaves no room for alternatives. This abstraction shows the influence of meta-theories in project management research. A meta-theory is a theoretical construct or paradigm portraying generic and reflexive properties that provide an avenue for scholars to question established assumptions [30,97]. Meta-theories were able to add new insights to project management in recent years, and have led to the formulation of a new branch of research along the lines of "project-based organising" [98,99]. The field of meta-theory has broadened its base in project management to include reflexive meta-theories, such as structuration theory [30,100,101], organisational learning theory [102,103], and practice theory [104].

## 5.2. Contrasting Definitions of Error and Lie Through Institutional Variation

The theoretical and methodological validity of Flyvbjerg, Holm, and Buhl's [4] reference to strategic misrepresentation and optimism bias formulations was claimed to lack any variation on the institutional variable pertaining to the planner's motives and rationality (Love and Ahiaga-Dagbui [5]). This was supported by Osland and Strand [105], who felt Flyvbjerg, Holm, and Buhl [4] applied the logic of suspicion in advancing the deduction that inaccurate cost forecasting came from optimism bias. They were critical of the wider school of thought, presenting an alternative perspective of project actors lying to the researchers.

Love and Ahiaga-Dagbui [5] argue that the word "lie" carries a weight of misinformation and jeopardises the status of the planners and the profession as a whole. It further projects harsh consequences and litigation proceedings, which will deter managers from PM. There must be some incentive and ethical consideration to attract scholars to the field of study. The brazen effrontery to litigate and accuse dedicated planners, and indirectly the engineering profession, of being liars in an industry built on uncertainty is a bold position. The conceptual formulation of the word lie is problematic in the context of PM, as it connotes the formulation of misleading statements with the deliberate intent of deception. However, this narrow view by [4], with sparse evidence linking "lie" with cost underestimation, is referred to as encapsulation [30].

## 5.3. Encapsulation

Arguably, scholars aligned with institutional frameworks are occupied with specialisation, and narrow conceptualisation referred to as encapsulation [30]. Encapsulation is an unintentional mechanism, but a subtle device that gives the illusion of sharing empirical domains and related vocabulary of mutual interest in the topic; however, it hinders rather than promotes cross-fertilisation [30]. Although it seems to demonstrate the successful co-joining of concepts [22], it tends to promote a limited and more restricted view of shared concepts [30].

There is a long-established gap between conventional project management theories that follow the deterministic path of on time, within budget, and sticking to a schedule,

and the indeterministic school that follows innovation management [30]. In their rebuttal, Flyvbjerg et al. [6] put forward their presentation by attributing these root causes to psychological and political bias. They envisage the problem as a risk. However, root causes can also be envisioned as uncertainty, which is more about innovative management (see Davies, Manning, and Söderlund [30]) rather than the deterministic setting of project management—on time, within budget, and according to specification [12,13]. Yet, [6] argued:

> *"Recent developments in behavioural science are causing Kuhnian paradigm shifts in many fields, including project management and forecasting. Love and Ahiaga-Dagbui [5] are on the wrong side of this shift."* (p. 183)

The above statement theorises that the cross-fertilisation of behavioural science with evidence science is normal and taken as a matter of course, a foregone acceptable way of doing things. This has become a habitual trait of the proponents of behavioural science to make this assumption. However, the acceptance of this assumption between two contrasting disciplines is unresolved. The Kuhnian paradigm shift and the acquisition of a technical name are insufficient to fill this gap between parallel aligned disciplines, as illustrated in Figure 1.

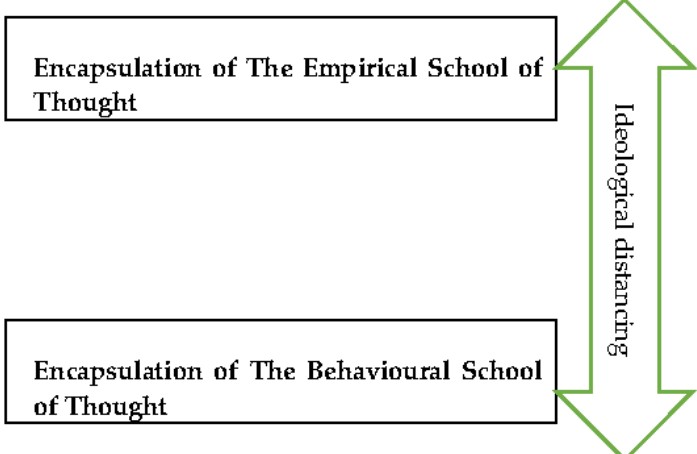

**Figure 1.** Visualisation of the parallel upbringings of cost overrun concepts and theories.

Flyvbjerg et al. [6] asserts, through the behavioural SOT, a shift to the Kuhnian paradigm. To support this theoretical conflation, Nobel prize winner psychologist Kahneman wrote:

> *"The treatment for the planning fallacy has now acquired a technical name, reference class forecasting, and Flyvbjerg has applied it to transportation projects in several countries."* [106]

The identification of psychological and political root causes, from among all causes of cost overruns, as a Kuhnian paradigm shift opens the gate from evidence science to admit behavioural science—from deterministic project management to indeterministic innovation management [86]. However, the theory of encapsulation submits that this is a narrow view of pretending to be aligned. Further, an ontological conceptualisation is necessary to accommodate these two contrasting theories purporting to decide between causes and root causes.

Central to the current theoretical arguments is understanding how the roots of the different ideological and epistemic concepts can infuse and converge. The current methodology of conflating concepts based on adhocracy has been exhausted in project management episteme. Flyvbjerg et al.'s [6] defence is as follows:

> *"In behavioural terms, the causal chain starts with human bias which leads to underestimation of scope during planning which leads to unaccounted for scope changes during*

*delivery which leads to cost overrun", explicitly stating, "Your biggest risk is you ... "* (p. 183)

The recognition of incoherent systems functioning together on infrastructure projects, and project management as a whole, requires modification of project praxis to curb cost overruns. Knowledge of psychological biases among competing stakeholders within the project environment is key to project success. The current methodology in project management is to conflate various theories from different disciplines into an incoherent projectised system for application, leading to the ineffective use of theories, which leads to inefficient project praxis, ultimately leading to the cost overrun phenomenon. Viewing the project system through co-joining efforts such as causal chain is a positive step towards theoretical advancement; little effort is made to cohere these diverse fields, which are left to operate as competing entities within the system. This is a classical case in which a multitude of scholars attempt to co-join the theoretical fundamentals of their respective disciplines, building on project management as a discipline. In this noble pursuit, we propose a reframing of the current cost overrun theoretical trajectory, a divergence from the fragmented approach of adhocracy and conflation to co-joining through transitions and causal linkages. Figure 2 proposes a simple visualisation of the proposed theoretical framework to reduce the current ideological distancing and encapsulation within cost overrun root causes research:

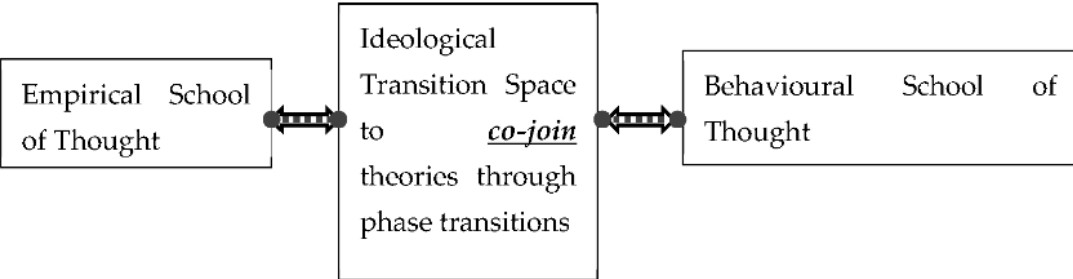

**Figure 2.** Theoretical conceptualisation for removing parallel theoretical upbringings through causal linkages as a co-joining mechanism.

The above theoretical framework recognises the two major schools of thought that govern the operations and analyses of cost overruns in project management. The recognition of the lack or non-existence of any formal mechanisms or affirmative expression of interest of how to fuse cognitive thinking, asserted in Flyvbjerg' s quotation above, into a concretised mechanistic established discipline, and respecting the different epistemic cultures, leads to the creation of a project ideological space with transitional boundaries. This ideological space leads to theory building through:

1. The removal of parallel upbringing of theories through encapsulation;
2. Bridging the gap created by the existing ideological divide between the dominant SOTs in cost overrun research;
3. Removing conflating concepts, which have yielded little positive results, into an organised co-joining relationship.

Simultaneity of theoretical applications, through causal linkages, with no pseudo-dichotomy, over transitionary boundaries, provides a workable framework to improve the cross-fertilisation of ideas across doctrines with diverse epistemological backgrounds.

## 6. Conclusions

This critical review paper aims to expose the growing divide among project management scholars in cost overrun research, based on their schools of thought rooted in either empirical sciences or behavioural sciences. Scholarly arguments followed the lines of the two main theoretical traditions concern with cost overruns: the first tradition has intellectual roots in engineering science and applied mathematics, primarily concerned with

the planning techniques and conventional mechanism of on time, within budget, and to a fixed schedule. The other tradition has its intellectual roots in the social sciences, such as sociology, organisation theory, and psychology, especially interested in the organisational and behavioural aspects of project organisations [2]. The feud ignited when behavioural scientists concluded that the cost overrun phenomenon is not technically rooted but resides mainly in political and psychological explanations; delusion (optimism) and deception (strategic misrepresentation) are the dominant explanations.

We discussed the expansion of the cost overrun phenomenon using three concepts: the pseudo dichotomy in logic, contrasting definitions of cost overruns, and encapsulation. These concepts led to the conclusion that the schools of thought developed separate and parallel identities to one another and, with the current trajectory, will not converge to share ideas if the current research trajectory is maintained. Firstly, we acknowledged that no universally accepted definition for cost overruns exists. We showed, using a literature case study approach, that explanations for root causes of cost overruns ignited a contentious debate among scholars, leading to the particular journal editors calling for a rejoinder in 2019. The focus was on the fundamental theoretical and empirical projections of the two main disciplines—the engineering tradition and the social science tradition—which are incompatible on cost overrun issues, for one avoids uncertainty to achieve determinateness, while the other assumes uncertainty and indeterminateness [2]. The call for a rejoinder was met with further episteme divergence and confrontational critical analysis.

Secondly, we showed scholars of the empirical school claimed the delusion and deception explanations create a false dichotomy, leading to the rejection of complexity in complex decision making. Subsequently, the empirics called on stakeholders and policymakers to revert to the original format of adopting a balanced approach through project characteristics such as engineering, management, complexity, geography, and politics. Scholars from the behavioural school of thought used the rejoinder articles by [5,54] as a point of departure from good practice. The nature of these explanations as root causes of cost overruns was strongly rebuked by the engineering and empirical scholars, citing that such explanations weaken linkages in decision making and bring the entire profession into disrepute. Contrary to Flyvbjerg's view that political and psychological explanations are the true root causes of cost overruns, we have shown that the intellectual roots are distinctly different, and expose a gap in the causal chain link between main roots causes and mainstream sub roots causes of cost overruns. From the case study, the journal's editors concluded that the parties failed to converge, suggesting there is room for further research. We have also exposed in the current debate that a theoretical causal link has not been established anywhere in the literature, and no attempts are made to co-join the two schools of thought. Ultimately, an ideological divide was created, expanding the knowledge gap in project cost overrun research. In attempting to return convergence in cost overrun research and aligning to the "third wave" of project management research [107], where single project-based theories are unified, this contemporary debate was reviewed through several perspectives.

Thirdly, the process of integrating concepts in this ad hoc manner originated from what is termed ideological distancing and encapsulation—previous studies indicating that diverse ideologies create distance through the incommensurable use of various languages [51,108]. Encapsulation is an unintentional mechanism, but a subtle device that gives the illusion of sharing empirical domains and related vocabulary of mutual interest in the topic; however, it hinders rather than promotes cross-fertilisation. To remove the pseudo-dichotomous stance, encapsulation, and ideological distancing, we propose the cross-fertilisation of ideas through the creation of project ideological space with transitional boundaries.

This paper has multiple implications for future research: it introduces a debate of historical interest among prominent scholars on cost overruns for the relatively young field of project management. The rejoinders requested by the editors in 2019 demonstrate further epistemological fissures in cost overrun research, leaving room for theoretical advancement

towards a unified theory into project management. This is of immense importance to the execution of large-scale and mega projects, where the phenomena of cost overruns have been resistant to theoretical and practical applications.

**Author Contributions:** Conceptualisation, A.A.C.; Methodology, A.A.C.; Validation, A.A.C.; Formal Analysis, A.A.C. and H.H.M.; Resources, A.A.C. and X.T.C.; Data Curation, A.A.C.; Writing—Original Draft Preparation, A.A.C.; Writing—Review and Editing, A.A.C., X.T.C., and H.H.M.; Visualisation, A.A.C.; Supervision, I.R. and A.M.; and Project Administration, A.A.C. All authors have read and agreed to the published version of the manuscript.

**Funding:** This research received no external funding.

**Institutional Review Board Statement:** Not applicable.

**Informed Consent Statement:** Not applicable.

**Data Availability Statement:** The data presented in this study are available on request from the corresponding author.

**Acknowledgments:** This study forms part of a Ph.D. research at the University of the West Indies, St. Augustine Campus, Trinidad and Tobago.

**Conflicts of Interest:** The authors declare no conflict of interest.

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
