# Peer review of "When Parallel Schools of Thought Fail to Converge: The Case of Cost Overruns in Project Management"

_buildings, doi:10.3390/buildings11080321_

Round 1

Reviewer 1 Report

Congratulations to the authors. Well structured and explained. A classic IMRaD structure implemented. Future research trends explained... Very good.

The only thing I would improve is the abstract which needs more concrete results. Especially the case that the authors have shown results which are opposite of some famous Authors.

Reviewer 2 Report

The paper analyses the project overrun phenomenon. It provides a thorough literature review of the reasons for this overrun and claims (repeatedly, though all the article) that there is a prominent "ideological distancing" between the two theories. Furthermore, the authors also claim (again, repeatedly) that there is a need to "integrate" the contrasting ideologies.
I would like to express my appreciation regarding the herculean work the authors went through in preparing this paper and it has a lot of merits in presenting the cost overrun phenomenon quite vividly.
However, in my opinion, the main hypothesis of the paper (the "conflict" or "ideological" distancing) is a bit farfetched.  Many phenomena have more than one root cause, and this one is no different. The claim that there is some "ideological" quarrel is a bit odd. Different Researchers may have different opinions on any subject and there may be several causes, it doesn't make it "ideological". Furthermore, the paper elaborates on the need for a "meta-theory" but fails to provide one.
The extensive work invested in this paper is important and it would be a shame if not published, therefore I suggest that the authors change the target of the paper (and its title, of course) and transfer it to a review of reasons for the overrun phenomenon. I am certain that there are more reasons than the two mentioned (e.g. Goldratt's theory of constraints for PM provides some insights), and it would be important to see a review of them all (and much of the current analysis may be used).
Minor issue: both figures have little to no contribution and just add unnecessary repetition to the already abundant claim.
I would like to see an amended paper, as this phenomenon deserves it.

Reviewer 3 Report

The paper is extremely informative and successfully presents the historical development of different school of thoughts regarding the cost overun of infrastructure project. 

It is suggested to improve the paper with a substantial numbers of infographics and timelines to highlight the methodology and the results of the study. 

Please check the title number of discussion and the conclusion: both 5.

Round 2

Reviewer 2 Report

The authors ignored most of my previous comments. I, therefore, cannot recommend accepting the paper as is. Please refer to my previous comments. 

Author Response

Dear Reviewer,

Round 3

Reviewer 2 Report

I still recommend that the work implemented in this article will be used for a comparison of various approaches (and not conflicts) to the problem of overrun.